# KD-Net: Continuous-Keystroke-Dynamics-Based Human Identification from RGB-D Image Sequences

**DOI:** 10.3390/s23208370

**Published:** 2023-10-10

**Authors:** Xinxin Dai, Ran Zhao, Pengpeng Hu, Adrian Munteanu

**Affiliations:** 1Department of Electronics and Informatics, Vrije Universiteit Brussel, 1050 Brussels, Belgium; xinxin.dai@vub.be (X.D.); ran.zhao@vub.be (R.Z.); adrian.munteanu@vub.be (A.M.); 2Centre for Computational Science and Mathematical Modelling, Coventry University, Coventry CV1 5FB, UK

**Keywords:** keystroke dynamics, human identification, RGB-D images, image sequences

## Abstract

Keystroke dynamics is a soft biometric based on the assumption that humans always type in uniquely characteristic manners. Previous works mainly focused on analyzing the key press or release events. Unlike these methods, we explored a novel visual modality of keystroke dynamics for human identification using a single RGB-D sensor. In order to verify this idea, we created a dataset dubbed KD-MultiModal, which contains 243.2 K frames of RGB images and depth images, obtained by recording a video of hand typing with a single RGB-D sensor. The dataset comprises RGB-D image sequences of 20 subjects (10 males and 10 females) typing sentences, and each subject typed around 20 sentences. In the task, only the hand and keyboard region contributed to the person identification, so we also propose methods of extracting Regions of Interest (RoIs) for each type of data. Unlike the data of the key press or release, our dataset not only captures the velocity of pressing and releasing different keys and the typing style of specific keys or combinations of keys, but also contains rich information on the hand shape and posture. To verify the validity of our proposed data, we adopted deep neural networks to learn distinguishing features from different data representations, including RGB-KD-Net, D-KD-Net, and RGBD-KD-Net. Simultaneously, the sequence of point clouds also can be obtained from depth images given the intrinsic parameters of the RGB-D sensor, so we also studied the performance of human identification based on the point clouds. Extensive experimental results showed that our idea works and the performance of the proposed method based on RGB-D images is the best, which achieved 99.44% accuracy based on the unseen real-world data. To inspire more researchers and facilitate relevant studies, the proposed dataset will be publicly accessible together with the publication of this paper.

## 1. Introduction

Keystroke dynamics is a soft biometric, which is widely used in many applications such as user authentication [1,2,3], fraud detection [4], biometric identification [5,6] and human–computer interaction [7]. Keystroke dynamics refers to the habitual patterns or rhythms an individual exhibits while typing on a keyboard input device. Existing studies mainly depend on the key press or release events [8], which focus on recording the velocities of pressing and releasing different keys [9], the typing styles of specific keys or combinations of keys [10], or the pressure exerted when pressing a key [11]. However, the study of vision-based keystroke dynamics is under-researched. With the development of RGB-D sensors, it is inexpensive to acquire the RGB and depth image sequences, which not only can capture the velocity of pressing and releasing different keys, but also contain rich information on the hand shape and typing posture. In this paper, we explored a new modality of keystroke dynamics for human identification from RGB-D image sequences.

Researchers have proposed various biometrics to validate human identity. These methods can be classified into two major categories according to the nature of the biometric characteristic used: hard biometrics and soft biometrics. Hard biometrics, also known as physiological biometrics, primarily involves physical characteristics that are intrinsic and unique to each individual, such as fingerprints, irises, DNA, etc. Hard biometrics is generally considered to be highly accurate and cannot be easily replicated and changed. However, such methods usually require physical contact with users, which can be uncomfortable and invasive. Moreover, the collected hard biometrics may be stolen and hacked, leading to risks of identity theft and fraudulent activities. In contrast, soft biometrics have boosted the interest of researchers and industry because of their ease of use, transparency, and a large number of potential applications [12]. Soft biometrics mainly includes signatures [13], gaits [14], keystrokes [15], etc. Signature biometrics is based on the strength of the signature, analyzing the movement of the pen, such as acceleration, pressure, and direction, and the length of the stroke, instead of the image of the signature. Such methods are easily influenced by the temperament and lifestyle of an individual. Gait biometrics identifies individuals by their stride length, footfall patterns, and other features of their gait. This dataset can then be used to create a unique biometric profile for the individual, which can be used for identification purposes.

In this paper, we mainly focused on RGB-D-based keystroke biometrics for two reasons: Firstly, we considered the diverse range of industrial environments where users might wear protective gear, thereby obscuring their facial features. Secondly, privacy is a critical concern in the digital landscape. Some individuals may be reluctant to use facial recognition due to concerns about data privacy and surveillance. Most existing methods [16,17,18] mainly depend on spatiotemporal RGB and depth images. However, they ignore cross-modal interaction. Our paper not only recovers the decoupled feature of the spatial domain and temporal domain, but also captures the cross-model feature of RGB and depth images.

The main contributions of this paper can be summarized as follows:We explored a novel visual modality of keystroke dynamics for human identification from RGB-D image sequences captured by a single RGB-D sensor.We created a novel dataset dubbed KD-MultiModal (RGB/depth images of keystroke dynamics) that contains 243.2 K frames of RGB images and depth images, obtained by recording a video of hand typing via an Azure Kinect. We will make the dataset public when the paper is published to facilitate related studies.We trained a deep neural network to verify the proposed modality of keystroke biometrics.

The rest of the paper is organized as follows. Section 2 briefly reviews related works that involved keystroke dynamics and gesture recognition. Section 3 describes the proposed data and adopted method. Section 4 presents the experimental results. The paper is concluded in Section 5.

## 2. Related Work

### 2.1. Keystroke Dynamics

Identifying humans based on keystroke dynamics has gained increasing attention as a forensic tool in soft biometrics. The existing keystroke biometric systems can be divided into two categories: fixed text [19] and free text [9]. Fixed text represents that the type of subject of the keystroke sequence is prefixed such as a password or username. Researchers [8,20] have studied keystroke biometrics based on fixed text to identify humans and the results can reach an accuracy rate of more than 95%. However, the application scenarios of such methods are severely limited by the text content. Free text [9] denotes the type of subject of the keystroke sequence, which can be any text, such as typing a sentence or writing an email. Murphy et al. [21] applied Gunetti and Picardi’s algorithm based on the free text keystroke dataset for authentication, which can achieve a classification error of 10.36%. However, it is far from the performance reached in the fixed text methods. Monaco et al. [22] proposed an algorithm based on statistical models, which showed that the performance using this approach in free text is close to that of fixed text, but it requires more keystroke data. Acien et al. [9] proposed a deep learning architecture for authentication and identification from 136 million keystrokes. Although it has the capability to achieve outstanding performance, it mainly relies on acquiring key press and release events, from which the following features can be extracted: interval, dwell time, latency, flight time, and up to up [5]. These features are easily affected by the velocities of typing, which are unstable and able to be forged. Moreover, recording these features requires keeping special software online, but the device usually shuts down, causing the software to go offline. More importantly, These features lack visual information, such as hand shapes and typing poses, which are important distinguishing characteristics of an individual. Therefore, we propose a visual modality of keystroke dynamics for human identification.

### 2.2. Hand Gesture Recognitions

A similar work to ours is hand gesture recognition, which also captures the movement of the hand as the input. Vision-based hand gesture recognition methods can be classified into RGB- [23,24,25,26], depth- [23,24,25,26], RGB-D- [23,24,25,26], and point-cloud-based gesture recognition [27,28] according to the type of input. These methods focus on effectively extracting spatiotemporal features for gesture recognition. For instance, Min et al. [27] leveraged PointLSTM to learn long-term spatiotemporal relationships from point cloud sequences. According to the experimental results, the accuracy of the point-cloud-based method is higher than the depth-based method and RGB-based method since the point cloud can precisely describe the potential geometric structure and distance information of the hand surface. Researchers proved that the recognition performance can be improved by employing multi-modal data. Abavisani et al. [23] leveraged the knowledge from multiple modalities to train unimodal 3D-CNNs for hand gesture recognition. Andrea et al. [24] proposed a new neural network architecture for hand gesture recognition, which embedded transformer layers to extract frame-level features and aggregate the temporal features. Yu et al. [25] proposed two-stage NAS to find a rate-related and modal-related network architecture, respectively, and optimal multi-modal feature transmission paths for RGB-D data. Zhou et al. [26] paid attention to the interaction of multi-modal features in the two independent dimensions of space and time. Following the trend that RGB-D-based methods usually perform better than RGB-based methods, we designed our model by taking RGB-D images as the input. Specifically, we selected the state-of-the-art method [26] as our backbone.

## 3. Methodology

### 3.1. Proposed Dataset

In order to verify our idea, a suitable dataset is necessary to train the neural network. Specifically, such a dataset should consist of keystroke sequences of RGB and depth images and ground-truth labels. Unfortunately, none of the existing public keystroke datasets met our requirements since no existing visual methods attempted to implement human identification from the keystroke. To address this problem, we created a novel keystroke dataset KD-MultiModal including RGB images, depth image sequences, and identity labels.

#### 3.1.1. Hardware Setup and Data Acquisition

To validate the proposed idea, we adopted an Azure Kinect sensor to collect data, Azure Kinect is developed by Microsoft Corporation, an American multinational technology corporation headquartered in Redmond, Washington. Specifically, an Azure Kinect sensor was first calibrated, and then, it was placed around 0.5 m above the keyboard to collect the data of the keystroke dynamics, as shown in Figure 1. The dataset involved 20 subjects (10 males and 10 females), who engaged in keyboard-intensive work on a daily basis. Each subject was asked to type around 20 sentences selected from the TypeNet dataset [9], which is the largest existing free text keystroke dataset. In order to make the typing scene as realistic as possible, the selected sentences widely covered daily words, including uppercase letters, lowercase letters, numbers, punctuation marks, and spaces. To conserve storage space, it took approximately 2 min for each subject to type a sentence. In total, our keystroke dataset contained 243.2K frames of RGB images and depth images, respectively. To compare the proposed method against relevant methods, the depth images were directly converted to point clouds given the intrinsic parameters of the sensor. Figure 2 shows examples of the proposed raw data directly captured from the Kinect.

#### 3.1.2. Data Preprocessing

##### RGB Image Preprocessing

As Figure 2 shows, only the hand and keyboard region contributed to the person identification. Therefore, the hands and keyboard need to be detected and segmented as Regions of Interest (RoIs). However, the segmentation of RoIs is challenging since the location of the keyboard may be moving or fixed and the location of the hand moves with the location of the keyboard. To track the location of the hand and keyboard, we first adopted Detectron2 [29] to detect the location of the hand. Then, the location of the keyboard was further determined based on the location of the hand and the keyboard size. Such a method is suitable for both a moving keyboard and a fixed keyboard since Detectron2 [29] is able to accurately detect the location of the hand, asshown in Figure 3. Specifically, Let $\left[B_{left}^{T},B_{top}^{T},B_{right}^{T},B_{bottom}^{T}\right]$ represent the bounding boxes captured by Detectron2, where $B_{left},B_{top},B_{right},B_{bottom}$ denote the pixel coordinates of the left, top, right, and bottom of the hand bounding boxes, respectively, and *T* indicate the type of bounding boxes as follows: R:Righthand, L:Lefthand ,K:Keyboardandhand. Our insight was that the movement range of the left hand is relatively small and mostly on the left side of the keyboard, so we took the left hand as a reference. By calculating the ratio of the pixels of the left-hand sideto those of the keyboard size, we were able to ascertain that the height of the keyboard is approximately the height of the hands, while the width of the keyboard is approximately 3.5-times wider than the width of the hand, so the keyboard size can be defined as:(1)$$\begin{array}{c} K_{w}=Round\left(3.5\times \left(B_{right}^{L}-B_{left}^{L}\right)\right)\\ K_{h}=Round\left(B_{bottom}^{L}-B_{top}^{L}\right) \end{array}$$
where $K_{w},K_{h}$ are the width and height of the keyboard, respectively. To prevent inconsistent errors in the numerical types, the pixels were treated as integers during the calculation process, which denotes a Round operation. The center of the left hand can be obtained by:(2)$$\begin{array}{c} C_{x},C_{y}=Round\left(\left(B_{right}^{L}+B_{left}^{L}\right)/2\right),\\ Round(\left(B_{bottom}^{L}+B_{top}^{L}\right)/2) \end{array}$$
Then, the bounding box that includes the hand and keyboard (i.e., RoI) is calculated based on the center of the left hand and keyboard size:(3)$$\begin{array}{c} B_{left}^{K}=C_{x}-K_{w}\\ B_{top}^{K}=C_{y}-K_{h}\\ B_{right}^{K}=C_{x}+Round\left(\frac{1}{4}K_{w}\right)\\ B_{bottom}^{K}=C_{y}+K_{h} \end{array}$$

##### Depth Image Preprocessing

Similar to RGB image prepossessing, it is also necessary to extract the RoI of the depth image. Specifically, we first removed irrelevant or noisy data from the depth images by setting the depth pixels greater than the mean pixel $D_{mean}$ to 0.
(4)$$\begin{array}{c} \hat{D}=\left\{\begin{array}{c} D,\;otherwise\\ 0,\;D_{\left(i,j\right)}>D_{mean} \end{array}\right. \end{array}$$
where *D* is the raw depth image directly captured from the RGB-D sensors. $\hat{D}$ is the denoised depth image. Then, we further extracted the RoI $\bar{D}$ of $\hat{D}$ following similar processing as for the RGB images. In the end, we employed a morphological operation to enhance the contour of the hand shape, which can be defined as:(5)$$\begin{array}{c} \ddot{D}=dilate\left(\bar{D},\left(k\times k\right)\right)-erode\left(\bar{D},\left(k\times k\right)\right) \end{array}$$
where dilate(·,(k×k)) and erode(·,(k×k)) indicate the dilation and erosion operations with a kernel size of k×k and $\ddot{D}$ is the final depth image. The process is shown in Figure 4.

##### Point Cloud Preprocessing

To explore the performance of different data representations in our study, we also took point clouds into consideration, which offer intuitive 3D information compared to RGB and depth images. Point clouds can be easily converted from depth images given the intrinsic parameters of RGB-D sensors. Each point in the cloud represents a specific location in 3D space and may also have additional information associated with it, such as color or intensity. We extracted the RoI starting with the *D* instead of $\ddot{D}$ since $\ddot{D}$ still contained irrelevant information such as a desk. The sequence of point clouds converted from the sequence of *D* will be the input of RoI extraction, as illustrated in Figure 5. Firstly, we used a ground-detection operation to detect the points around the hand and keyboard and remove them.
(6)$$\begin{array}{c} {\hat{P}}^{n}=P^{n}-\Psi \left(P^{n},\epsilon_{1}\right) \end{array}$$
where $P^{n}=\left\{p_{i}^{n}\in R^{3}|i=1,2,\ldots ,I^{n}\right\}$ represents a set of points with $I^{n}$ points captured from the $n_{th}$ frame; $\Psi (\cdot ,\epsilon_{1})$ indicates the ground-detection operation with distance threshold $\epsilon_{1}$; ${\hat{P}}^{n}$ is the $n_{th}$ point cloud removed around the hand and keyboard. Then, we used the ${\left(n-1\right)}_{th}$ RoI to search the k-nearest neighbor points of $n_{th}$:(7)$$\begin{array}{c} {\bar{P}}^{n}=\left\{\begin{array}{c} P_{T},\;\;\;\;\;n=1\\ (\Phi {\left({\left({\hat{p}}_{k}^{n}\right)}_{k=1}^{K},{\ddot{p}}_{j}^{n-1},\epsilon_{2}\right)}_{j=1}^{J^{n-1}},\;n>1 \end{array}\right. \end{array}$$
where $P_{T}$ is the template of the keyboard and hand; $\Phi (\cdot ,\cdot ,\epsilon_{2})$ is the k-NN search operation with distance threshold $\epsilon_{2}$; ${\ddot{p}}_{j}^{n-1}$ is the $j_{\mathrm{th}}$ point of the ${\left(n-1\right)}_{th}$ extracted RoI ${\ddot{P}}^{n-1}$; ${\left({\hat{p}}_{i,k}^{n}\right)}_{k=1}^{K}$ are the *K* points of ${\hat{P}}^{n}$ searched by ${\ddot{P}}^{n-1}$. However, there were still some outlier points around the RoI. We employed the clustering method to divide ${\bar{P}}^{n}$ into several categories and keep the largest number of point clouds to obtain the RoI ${\ddot{P}}^{n}$.

### 3.2. Human Identification from Visual Keystroke Dynamics

The key idea of this study was to explore the possibility of recognizing humans from keystroke dynamics RGB-D image sequences. We took the state-of-the-art work [26] of motion recognition as the backbone. We first introduce the method and then design the experimental strategy. The method is RGB-D-based multi-modal spatiotemporal representation learning for human identification, which includes four modules: Decoupling Spatial Feature learning (DSF) module, Decoupling Temporal Feature learning (DTF) module, Spatiotemporal Recoupling (STR) module, and cross-modal interactive learning module (CAPF). Regarding the experimental strategy, this method can also be applied to unimodal branches (i.e., RGB-based modal and depth-based model), so we split the method into two branches to explore the effect of different features on the recognition performance. The results will be shown in Section 4.1.

#### 3.2.1. Decoupling Spatiotemporal Feature Module

To extract the spatial and temporal features from the sequence of images, we adopted two modules: Decoupled Spatial Feature learning (DSF) and Decoupled Temporal Feature learning (DTF). Specifically, let $I=[I_{1},I_{2},\cdots ,I_{T}]$ denote the input with *T* sampled from the video. It is firstly fed into the DSF to learn spatial feature O. The DSF consists of the SMS and FRP modules. The former is mainly responsible for extracting the multi-scale features *f*, while the latter guides each SMS module to focus on local important areas in the image to generate visual guidance maps G. Then, the spatial features can be obtained by:(8)$$\begin{array}{c} O=\left(f⊙G\right)⊗f+f,O\in R^{T\times d} \end{array}$$
where ⊙ and ⊗ represent the elementwise product and 1D convolution operation, respectively. The DTF takes the enhanced spatial features $\ddot{O}\in R^{T\times d/2}$ as the input, which is obtained in the spatiotemporal Recoupling Module (RCM) (as described in Section 3.2.2). $\ddot{O}$ first is sampled into a sub-sequence of features ${\ddot{O}}_{k}\in R^{T_{n}\times d/2}$ with length $T_{n}$, where $k=R\left[\left\lceil {\frac{T}{T}}_{n}\right\rceil \times t-1,\left\lceil {\frac{T}{T}}_{n}\right\rceil \times t\right],s.t.\;t=1,2,\cdots ,T_{n}$, $R\left[a,b\right]$ represents randomly selecting an integer. The sampled features ${\ddot{O}}_{k}$ are fed into the Temporal Multi-Scale feature-learning module (TMS) to extract the local fine-grained spatial features ${\ddot{O}}_{k}^{L}$. After that, they are fed into a stack of Transformer blocks to learn the coarse-grained temporal representation ${\ddot{O}}_{k}^{G}$. Furthermore, to avoid overfitting to one of the sub-branches, we introduced a temperature parameter τ to control the sharpness of the output distribution of each sub-branch and impose a constraint loss on it. Therefore, the output of the DTF network can be formulated as:(9)$$\begin{array}{c} O^{CLS}=\sum_{k=1}^{K}MLP\left(O_{k}^{CLS}\right)/\tau \end{array}$$
where $O^{CLS}$ is the class token vector embedded in the Transformer block and *K* indicates the number of subbranches. τ follows a cosine schedule from 0.04 to 0.07 during the training.

#### 3.2.2. Spatiotemporal Recoupling Module

In Section 3.2.1, we obtained the decoupled feature of the spatial domain and temporal domain. However, they are independent. To solve this issue, a recoupling strategy was applied to establish space–time interdependence, which can enhance the robustness of the spatial features by incorporating the time domain into the space domain. Based on spatial features *O* from the DSF and temporal features $O^{CLS}$ from the DTF, a Spatiotemporal Recoupling (STR) module is proposed to strengthen the space–time connection during training. Specifically, *O* is first linearly mapped into a low-dimensional space to obtain the mapped features $\bar{O}\in R^{T\times d/2}$. Then, it is transposed and fed into the RCM to enhance the spatial features from the *X* (feature dimension) and *Y* (sequence dimension) directions. To describe the correlation of the intra-frame in the sequence along the *X* direction, the self-attention mechanism [30] is introduced to calculate the attention map $A_{X}$:(10)$$\begin{array}{c} A_{X}=\delta \left(GAP\left(\frac{QK^{T}}{\sqrt{d}}\right)\right),A_{X}\in R^{1\times d/2}\\ s.t.\;Q={\bar{O}}^{T}W_{Q},K={\bar{O}}^{T}W_{K} \end{array}$$
where $W_{Q}$ and $W_{K}$ are the learnable weights of *Q* and *K*, which denote the queries and keys, δ is the Sigmoid activation function, and GAP indicates the adaptive Global Average Pooling operation. For the Y direction (inter-frame), the correlation of the inter-frame in the sequence can be described by:(11)$$\begin{array}{c} A_{Y}=\delta \left(MLP\left(GAP\left({\bar{O}}^{T}\right)\right)\right),A_{Y}\in R^{1\times T} \end{array}$$
Then, $A_{X}$ and $A_{Y}$ are combined with *O* by the elementwise product operation to enhance spatial feature $\hat{O}$:(12)$$\begin{array}{c} \hat{O}=O⊙\sum_{i=1}^{I}\sum_{j=1}^{J}A_{X,i}^{T}\times A_{Y,j} \end{array}$$
$\hat{O}$ is used for temporal representation learning, as introduced in Section 3.2.1.

#### 3.2.3. Cross-Modal Interactive Learning

As previously mentioned, RGB and depth images can form complementary features, so we employed two network branches to learn the spatiotemporal features of the RGB and depth images and interact with them in the spatial and temporal dimensions respectively. Take the spatial features as an example; ${\hat{O}}_{R}$ and ${\hat{O}}_{D}$ of the RGB-D modalities extracted from the STR first interact through the MLP layer to generate a joint spatial representation, which is further integrated with raw spatial features by a residual structure:(13)$$\begin{array}{c} {\ddot{O}}_{R}=LN\left(MLP\left(\left[{\hat{O}}_{R};{\hat{O}}_{D}\right]\right)\right)+O_{R}\\ {\ddot{O}}_{D}=LN\left(MLP\left(\left[{\hat{O}}_{R};{\hat{O}}_{D}\right]\right)\right)+O_{D} \end{array}$$
where ${\ddot{O}}_{R}$ and ${\ddot{O}}_{D}$ denote the fused spatial features of the RGB-D modalities, respectively; [;] represents the concatenation. Obtaining the fused temporal features ${\ddot{O}}_{R}^{CLS}$ and ${\ddot{O}}_{D}^{CLS}$ of the RGB-D modalities uses a similar processing to the interactive way of spatial features. They are the inputs for the Cross-modal Adaptive Posterior Fusion module (CAPF), which consists of an Encoder and a Decoder. The outputs of the Encoder and a Decoder represent ${\ddot{O}}^{E}$ and ${\ddot{O}}^{D}$, which can be used to supervise the network.

#### 3.2.4. Experimental Strategy

We took the method introduced above as the backbone, called KD-Net, to explore the feasibility of human identification from the novel visual keystroke dynamics. Figure 6a shows the architecture of RGBD-KD-Net, which follows KD-Net. RGBD-KD-Net takes the RGB-D image sequences of the keystroke dynamics as the input. The continuous sequences of images can capture the typing habits of the subjects. The RGB images contain texture information such as hand color. The depth images can describe the location and shape of the hand such as the distance between the hand and the keyboard and the size of the hand. All of the information is the discrimination characteristics for human identification. RGBD-KD-Net is able to establish the connection between them from the spatiotemporal and multi-modal aspects to identify humans. Moreover, we further explored the effect of different features on the recognition performance, so we extended RGBD-KD-Net to the RGB and depth image inputs, individually called RGB-KD-Net/D-KD-Net, as shown in Figure 6b. Compared with RGBD-KD-Net, RGB-KD-Net only takes RGB image sequences as the input and removes the step of RGB and depth interaction. Thus, it is only able to extract the RGB-based spatiotemporal representation. Such a representation lacks depth information from depth images. D-KD-Net is similar to RGB-KD-Net. It takes depth image sequences as the input and can only learn depth-based spatiotemporal representation, which lacks texture information from RGB images. We will discuss their performance in Section 4.

#### 3.2.5. Loss Functions

To train the KD-Net network, a multi-loss function was used; it is a weighted sum of three loss functions: $L_{d}$, $L_{c}$, $L_{b}$, and $L_{m}$, which are defined in Equations (Equation 14)–(Equation 17), respectively. $L_{d}$ is the similar in RGBD-KD-Net, RGB-KD-Net, and D-KD-Net. For $L_{c}$, $L_{b}$, and $L_{m}$, RGB-KD-Net and D-KD-Net use enhanced spatial features $\hat{O}$ and temporal features $O^{CLS}$ to supervise the network, while RGBD-KD-Net uses fused spatial features $\ddot{O}$ and fused temporal features ${\ddot{O}}^{CLS}$ to supervise the network. The following will introduce the loss functions in the RGBD-KD-Net.

The self-distillation loss function is introduced to enhance the inter-frame correlation between the temporal features and spatial features.
(14)$$\begin{array}{c} L_{d}={\frac{1}{N}}_{b}\sum_{i=1}^{N_{b}}KL\left(\frac{\tau O_{i}^{CLS}}{T}-\frac{F(MLP\left(GAP\left({\bar{O}}_{i}^{T}\right)\right))}{T}\right) \end{array}$$
where $N_{b}$ is the batch size, KL indicates the Kullback–Leibler divergence [31], F indicates the linear mapping function, and T is the distillation temperature parameter.

Cross-entropy loss is used to measure the classification performance on the RGB images, depth images, and encoder features, which can be formulated as follows: (15)$$\begin{array}{c} L_{c}=L_{c}\left(C\left({\ddot{O}}_{R}^{CLS}\right),GT\right)+L_{c}\left(C\left({\ddot{O}}_{D}^{CLS}\right),GT\right)+L_{c}\left(C\left({\ddot{O}}_{E}\right),GT\right)\\ s.t.\;L_{c}\left(y,p\right)=-{\frac{1}{N}}_{b}\sum_{i=1}^{N_{b}}\sum_{j=1}^{N_{c}}y_{ij}log\left(p_{ij}\right)) \end{array}$$
where $L_{c}$ is the cross-entropy loss, C is the linear classifier, GT is the ground-truth label, and $N_{c}$ is the number of classes. When $N_{c}=2$, GT is set to 0 or 1, and the cross-entropy loss is the binary cross-entropy loss $L_{b}$.

The mean-squared errors between fused temporal features and the decoded feature are used to supervise the encoder of the CAPF.
(16)$$\begin{array}{c} L_{m}=\frac{1}{|{\ddot{O}}_{D}|}\sum_{i=1}^{|{\ddot{O}}_{D}|}\left|\right|{\left({\ddot{O}}_{D}\right)}_{i}-{\left({\ddot{O}}_{R}^{CLS}\right)}_{i}{\left|\right|}_{2}^{2}\\ +\frac{1}{|{\ddot{O}}_{D}|}\sum_{i=1}^{|{\ddot{O}}_{D}|}\left|\right|{\left({\ddot{O}}_{D}\right)}_{i}-{\left({\ddot{O}}_{D}^{CLS}\right)}_{i}{\left|\right|}_{2}^{2} \end{array}$$
The weighted sum of these three loss functions is employed to train the network: (17)$$\begin{array}{c} L=\alpha_{1}L_{d}+\alpha_{2}L_{c}+\alpha_{3}L_{b}+\alpha_{4}L_{m} \end{array}$$

## 4. Results

Our collected dataset consisted of 243.2 K frames of RGB images and depth images from 20 subjects. Each subject typed 20 different sentences to obtain 1000 videos. We adopted the released leave-one-split-out cross-validation protocol, which divides the 20 subjects into 10 groups and uses nine splits for training and the rest for testing. The proposed method was implemented with Pytorch. The networks were trained for 35 epochs with a batch size of four. The optimizer employed SGD with a weight decay of 0.0003 and a momentum of 0.9. The learning rate was linearly ramped up to 0.01 during the first three epochs and then decayed with a cosine schedule. In the loss function, we set $\alpha_{1}=1,\;\alpha_{2}=1,\;\alpha_{3}=1,\;\alpha_{4}=1$. The input sequences were randomly/center-cropped into 224 × 224 during training/inference and were augmented by random clipping and rotating.

We did not have keystroke-related method comparisons because our data were not available for them. We compared the results from different data representations: RGB images, depth images, RGB-D images, and point clouds. The first three data representations were mainly based on the method proposed in Section 3.2, while the last one was conducted with the point-based state-of-the-art method PointLSTM [27].

### 4.1. Self-Comparison

In this section, we evaluate the model on three different types of datasets (RGB, depth, RGB-D images), which were introduced in Section 3.1. Table 1 compares their performance. It can be seen that RGBD-KD-Net had the best performance, while the results of RGB-KD-Net were the worst, which demonstrates that multi-modal data can provide more-robust discrimination features. Moreover, it is worth noting that RGBD-KD-Net had a 5.03% improvement compared with RGB-KD-Net and it improved by 1.87% on the accuracy compared with D-KD-Net, which shows that adding depth information can significantly improve recognition accuracy. Besides, the accuracy of RGBD-KD-Net reached 99.44%, which proves that the novel visual modality of the keystroke dynamics is effective for human identification.

Figure 7 presents confusion matrices based on the three different types of datasets. The results on the diagonal indicate classification accuracy between the predicted labels and the ground-truth labels, while the other results indicate that the predicted labels and the ground-truth labels are inconsistent, which means classification error. We observed that RGBD-KD-Net presented a well-diagonalized confusion matrix between the predicted labels and the ground-truth labels. The classification accuracy on the diagonal was close to 100%. Compared with the results of RGB-KD-Net, RGBD-KD-Net had 100% and 12% improvements on the 15th and 19th classes, respectively. In contrast, it had a 38% improvement on the first class compared with D-KD-Net, which demonstrates that combining the hand color provided by the RGB images and the hand shape provided by depth images helps to identify different hands and then identify people.

### 4.2. State-of-the-Art Comparison

As described in Section 3.1, point clouds can be directly converted from depth images. Therefore, we also used PointLSTM, a state-of-the-art method [27], to explore more data representations and to compare with RGBD-KD-Net. Besides, we also compared the proposed method with the state-of-the-art spatiotemporal method called STRM [32], which only considers the spatiotemporal relationship of RGB sequences. Their classification accuracy is shown in Table 2. We can see that RGBD-KD-Net performs better than STRM and PointLSTM, which further demonstrates that the hand color provided by the RGB images and the depth information from the depth images comprise valuable information in human identification. Furthermore, the coupled spatiotemporal relationship and cross-modal interaction can improve the recognition accuracy. To visualize the recognition accuracy of each class. we also present confusion matrices to compare the classification accuracy of each class, as shown in Figure 8. Compared with STRM, RGBD-KD-Net made significant improvements in the first two classes. We also observed that RGBD-KD-Net can improve slightly on the 2nd, 11th, and 19th classes compared with PointLSTM. We show the RGB images of these three subjects in Figure 9. We can see their hand colors are different; Subject 2 is the lightest; Subject 11 is the darkest. Therefore, RGBD-KD-Net can improve the recognition accuracy of these three subjects by considering the hand color information.

Besides, we also compared the proposed method with the state-of-the-art spatiotemporal method called STRM [32], which only considers the spatiotemporal relationship of RGB sequences. Their classification accuracy is shown in Table 2. We can see that RGBD-KD-Net performed better than STRM and PointLSTM, which further demonstrates that the hand color provided by the RGB images and the depth information from the depth images comprise valuable information in human identification. Furthermore, the coupled spatiotemporal relationship and cross-modal interaction can improve the recognition accuracy.

### 4.3. Evaluation of the Number of Frames

To evaluate the impact of different frame numbers on recognition accuracy, we conducted four experiments that took 4, 8, 16, and 32 frames as the input, respectively. The results are summarized in Table 3, from which we can see there was a substantial gain in the recognition accuracy as the number of frames increased from 4 to 16, with the accuracy soaring from 93.04% to 99.44%. However, a slight dip in accuracy was observed when transitioning from 16 frames to 32 frames, reaching 97.91%. This suggests a potential saturation point where additional frames do not yield significant accuracy improvements. Furthermore, the training time and inferencetime exhibited a proportional increase with the number of frames, but the jump from 16 to 32 frames resulted in a considerable rise in the training time, reaching 47.5 h and 2.9 s. Consequently, there appears to be an optimal frame number, 16 frames, that strikes a harmonious balance between high accuracy and reasonable training time. Nevertheless, the ultimate choice should be informed by the specific application’s requirements and resource constraints, considering the trade-offs between accuracy and computational costs.

## 5. Conclusions

In this work, we explored a novel visual modality of keystroke biometrics for human identification and studied the performance of different data representations. We created a dataset that contained 243.2 K frame RGB images and depth images of 20 subjects typing 20 different sentences and converting the point clouds from depth images. We compared the results on the RGB images, depth images, RGB-D images, and point clouds. The classification accuracy on the depth images was higher than that on the RGB images, which demonstrates that the features of the hand shape are more effective for distinguishing individuals than hand color. The results on the RGB-D images were better than the results on the RGB and depth images, which indicates that the hand color provided by the RGB images can be used as a supplement to the hand shape. Moreover, we also compared the results between the images and point clouds. The results on the point clouds were better than the results on the depth images, which shows that background information can affect classification accuracy. However, they were worse than the results on the RGB-D images, which further demonstrates that hand color is valuable information in human identity. In conclusion, the novel visual modality of keystroke biometrics can achieve high classification accuracy and multi-modal data can provide robust discrimination characteristics.

In the future, we will consider different keyboards and collect more hand typing data. Moreover, the method mainly relies on RGB-D images, which lack point cloud information. This limitation can be mitigated by fusing the information of the RGB images, depth images, and point clouds.

## Figures and Tables

**Figure 1 sensors-23-08370-f001:**
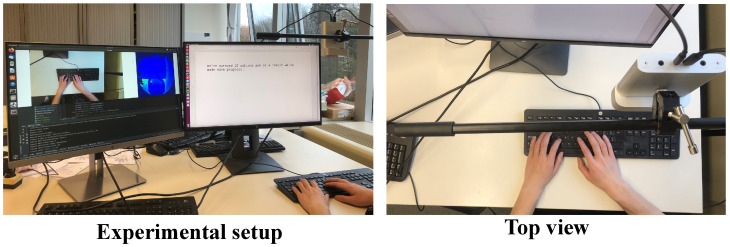
The setup of keystroke data collection.

**Figure 2 sensors-23-08370-f002:**
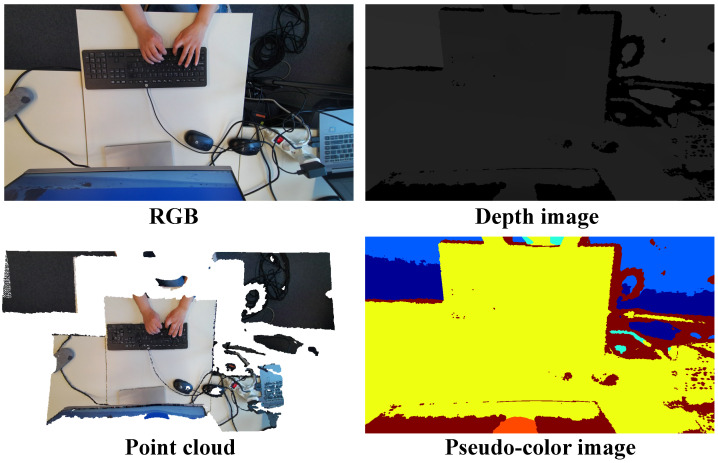
Examples of the collected keystroke data.

**Figure 3 sensors-23-08370-f003:**
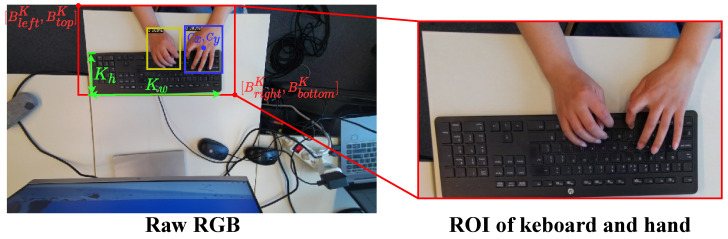
RoI extraction: the yellow rectangle and blue rectangle are the bounding boxes of the right hand and left hand captured by Detectron [29], which are denoted as $\left[B_{left}^{R},B_{top}^{R},B_{right}^{R},B_{bottom}^{R}\right]$, $\left[B_{left}^{L},B_{top}^{L},B_{right}^{L},B_{bottom}^{L}\right]$. $C_{x},C_{y}$ is the center of the left hand. $K_{w},K_{h}$ are the width and height of the keyboard. The red rectangle is the bounding boxes of the RoI $\left[B_{left}^{K},B_{top}^{K},B_{right}^{K},B_{bottom}^{K}\right]$.

**Figure 4 sensors-23-08370-f004:**
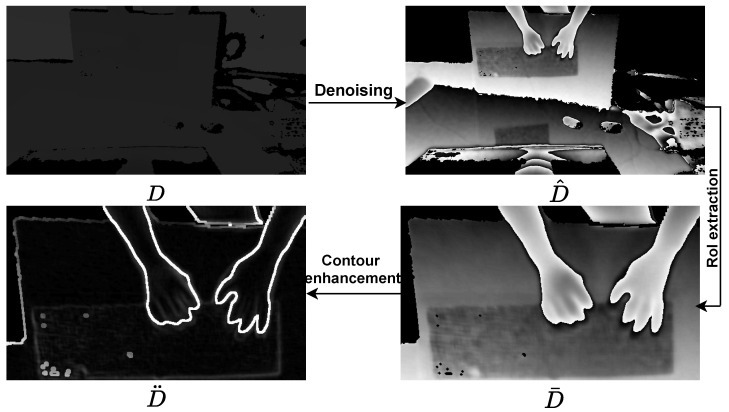
The processing of the depth image.

**Figure 5 sensors-23-08370-f005:**
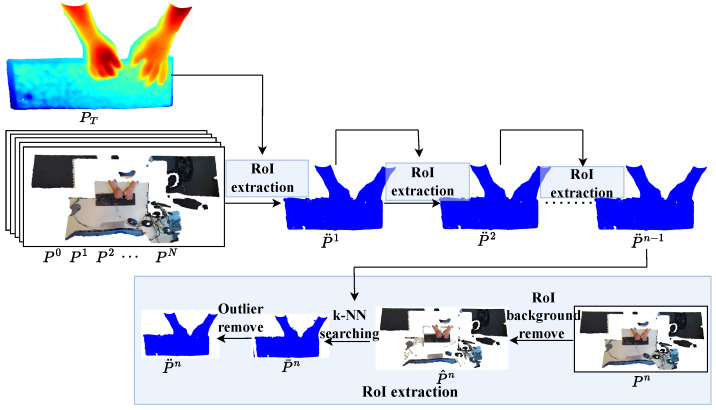
The process of RoI extraction for the point cloud sequence.

**Figure 6 sensors-23-08370-f006:**
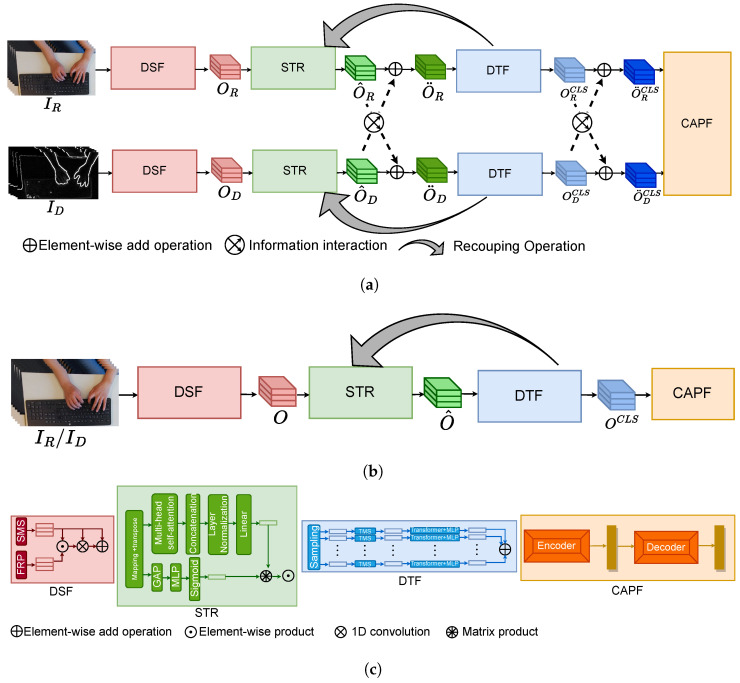
KD-Net with different inputs. It consists of four modules: the Decoupled Spatial Feature (DSF) module, the Decoupled Temporal Feature (DTF) module, the Spatiotemporal Recoupling (STR) module, and the Cross-modal Adaptive Posterior Fusion (CAPF) module. (**a**) The architecture of RGBD-KD-Net. It takes the RGB-D image sequences of the keystroke dynamics as the input and learns the coupled multi-modal spatiotemporal representation for human identification. (**b**) The architecture of RGB-KD-Net/D-KD Net removes the step of RGB and depth interaction. It takes the RGB/depth image sequences as the input and extracts the RGB-/depth-based spatiotemporal representation for human identification. (**c**) The details of the DSF, DTF, STR, and CAPF modules. The DSF module is responsible for learning spatial features *O*. The DTF module is responsible for learning temporal features $O^{CLS}$. The STR module is used for strengthening the space–time connection of *O* and $O^{CLS}$ to obtain enhanced spatial features $\hat{O}$. The CAPF module is used to conduct deep multi-modal representation fusion by an encoder and a decoder.

**Figure 7 sensors-23-08370-f007:**
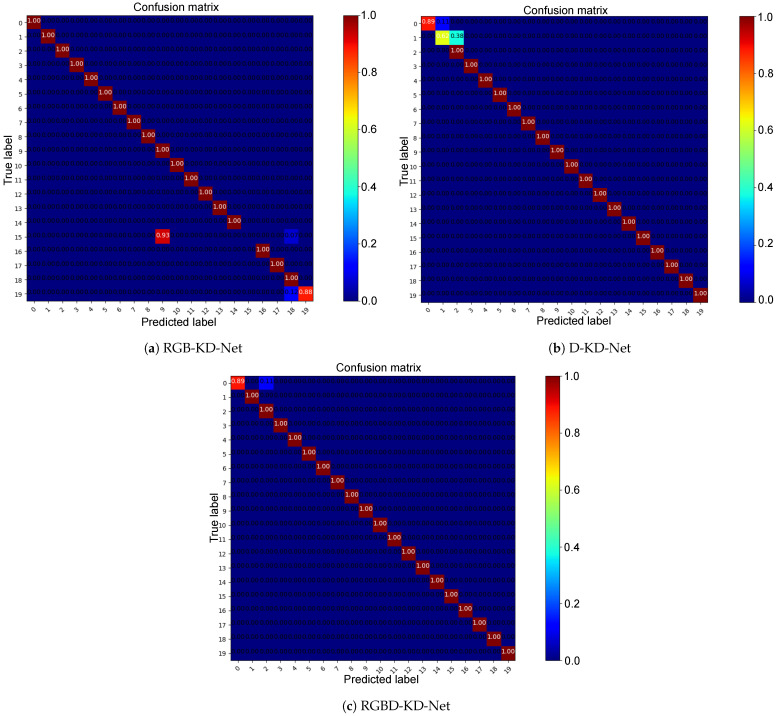
The confusion matrices were obtained by comparing the ground-truth labels and the predicted labels on RGB-KD-Net, D-KD-Net, and RGBD-KD-Net. Best seen on a computer with color and zoom.

**Figure 8 sensors-23-08370-f008:**
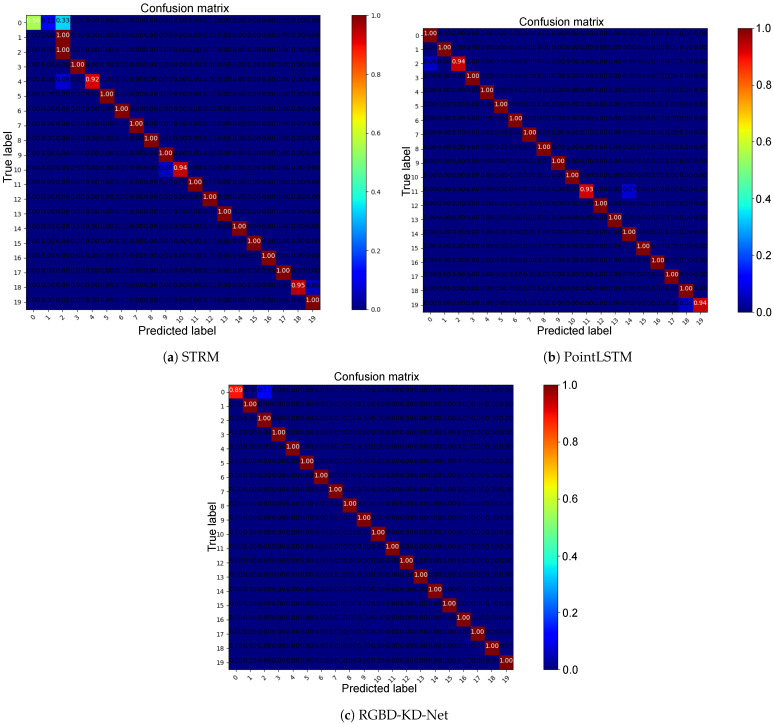
The confusion matrices obtained by comparing the ground-truth labels and the predicted labels on STRM, PointLSTM, and RGBD-KD-Net. It is best seen on a computer with color and zoom.

**Figure 9 sensors-23-08370-f009:**
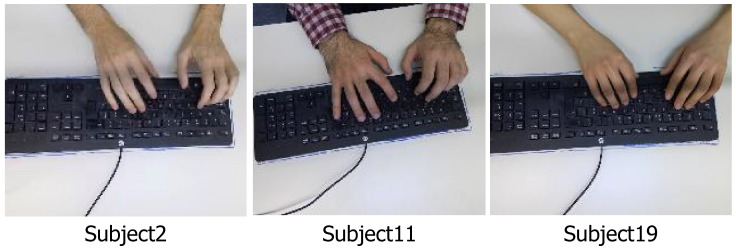
The RGB images of Subject 2, Subject 11, and Subject 19. They have different hand colors: Subject 2 is the lightest, and Subject 11 is the darkest.

**Table 1 sensors-23-08370-t001:** Comparison results with RGB-KD-Net, D-KD-Net, and RGBD-KD-Net.

Method	Accuracy (%)
RGB-KD-Net	94.41
D-KD-Net	97.57
RGBD-KD-Net	99.44

**Table 2 sensors-23-08370-t002:** Comparison results with STRM, PointLSTM, and RGBD-KD-Net. (Bold font indicates the highest recognition accuracy).

Method	Accuracy (%)
STRM [32]	91.85
PointLSTM [27]	98.00
RGBD-KD-Net	**99.44**

**Table 3 sensors-23-08370-t003:** Comparison results with the different number of frames of RGBD-KD-Net.

Number of Frames	Accuracy (%)	Training Time (h)	Inferencetime (s)
4	93.04	15.5	1.8
8	97.32	25.5	1.9
16	99.44	33.5	2.2
32	97.91	47.5	2.9

## Data Availability

The data are not availabledue to restrictions, e.g., privacy or ethical.

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
