# Peer review of "KD-Net: Continuous-Keystroke-Dynamics-Based Human Identification from RGB-D Image Sequences"

_sensors, 2023, doi:10.3390/s23208370_

Round 1

Reviewer 1 Report

The Authors must be ameliorate the describes and the clearly of methods, are very confused

Author Response

We would like to thank you for the positive evaluation of our work as well as for the detailed comments and suggestions that helped improving our paper. Please refer to the attached responses to your comment.

Reviewer 2 Report

[Strengths]

  The author explored a novel visual modality of keystroke dynamics for human identification from RGB-D image sequences captured by a single RGB-D sensor.

  Author created a novel dataset dubbed KD-MultiModal (RGB/depth-images of keystroke dynamics) that contains 243.2K frames of RGB images and depth images, obtained by recording the video of hand typing via an Azure Kinect. The author will make the dataset public when the paper is published to facilitate the related study.

  Author trained a deep neural network to verify the proposed modality of keystroke biometrics.

[Weaknesses]

  Existing studies mainly depend on the key press or release events, which focus on recording the velocities of pressing and releasing different keys, the typing styles of specific keys or combinations of keys, or the pressure exerted when pressing a key. The aforementioned work is based on restricted scenario assumptions, i.e., we don't have a camera scenario for authentication. If we have a camera, why don't we just do face recognition? What is the value of the application of identity recognition for hand-based information?

  The paper only compares its work to PointLSTM. It would be more helpful to compare the paper's work with more recent research.

  Learning spatiotemporal features is a crucial component of this model. However, there is a lack of an introduction to existing spatiotemporal feature learning methods. Please provide some methods for references, such as [1], [2], [3].

[1] Two-stream convolutional networks for action recognition in videos

[2] Towards Omni-Supervised Face Alignment for Large Scale Unlabeled Videos.

[3] Stm: Spatiotemporal and motion encoding for action recognition

Concerns about the writing. Please take a close look at the entire manuscript. For example, (1). ‘ROI of keboard and hand’ in Figure 3.

Author Response

(The authors gave the same response as above.)

Reviewer 3 Report

This paper discussed a visual modality of keystroke biometrics for human identification using a single RGB-D sensor. The authors create a dataset that contains 243.2K frames of RGB images and depth images, obtained by recording the video of hand typing with a single RGB-D sensor. Deep neural networks are adopted to learn features from different data representations, including RGB-KD-Net, D-KD-Net and RGBD-KD-Net. The results show that  RGB-D is more accurate in distinguishing individuals than RGB and point clouds. This work is interesting for using keystroke biometrics for human identification. There are two questions about the work.

1. For ROI extraction, there is a RGB-D sensor to capture the images, how do not use the deepth information to get the keybord and hand area directly?

2. For spetiotemporal sequences, how many frames is more suitable for the identification network.

Author Response

(The authors gave the same response as above.)

Round 2

Reviewer 2 Report

The author addressed all my concerns.